# Synthesis of High Crystallinity 1.13 nm Tobermorite and Xonotlite from Natural Rocks, Their Properties and Application for Heat-Resistant Products

**DOI:** 10.3390/ma15103474

**Published:** 2022-05-12

**Authors:** R. Siauciunas, G. Smalakys, A. Eisinas, E. Prichockiene

**Affiliations:** Department of Silicate Technology, Kaunas University of Technology, Radvilenu pl. 19, 50270 Kaunas, Lithuania; giedrius.smalakys@gmail.com (G.S.); anatolijus.eisinas@ktu.lt (A.E.); edita.rupsyte@ktu.lt (E.P.)

**Keywords:** high-crystallinity 1.13 nm tobermorite and xonotlite, heat-resistant insulating materials, hydrothermal curing

## Abstract

The main measure to reduce energy losses is the usage of insulating materials. When the temperature exceeds 500 °C, silicate and ceramic products are most commonly used. In this work, high-crystallinity 1.13 nm tobermorite and xonotlite were hydrothermally synthesized from lime and Ca–Si sedimentary rock, opoka. By XRD, DSC, TG and dilatometry methods, it has been shown that 1.13 nm tobermorite becomes the predominant compound in stirred suspensions at 200 °C after 4 h of synthesis in the mixture with a molar ratio CaO/SiO_2_ = 0.83. It is suitable for the production of insulating products with good physical–mechanical properties (average density < 200 kg·m^−1^, compressive strength ~0.9 MPa) but has a limited operating temperature (up to 700 °C). Sufficiently pure xonotlite should be used to obtain materials with a higher operating temperature. Even small amounts of semi-amorphous C–S–H(I) significantly increase its linear shrinkage during firing. It has also been observed that an increase in the strength values of the samples correlated well with the increase in the size of xonotlite crystallites. The optimal technological parameters are as follows: molar ratio of mixture CaO/SiO_2_ = 1.2; water/solid ratio W/S = 20.0; duration of hydrothermal synthesis at 220 °C—8 h, duration of autoclaving at 220 °C—4 h. The average density of the samples was ~180 kg·m^−1^, the operating temperature was at least 1000 °C, and the compressive strengths exceeded 1.5 MPa.

## 1. Introduction

The 1.13 nm tobermorite Ca_5_Si_6_O_16_(OH)_2_·4H_2_O, calcium silicate hydrate of potentially highest practical importance, is formed in autoclaved aerated concrete, sand-lime bricks, fiber cement roofing and facade products, heat-insulating materials, etc. Another important application stems from its high sorption capacity for heavy and radioactive metal ions [1,2,3]. The extensive application of 1.13 nm tobermorite is due to the differences in the physical behavior of the crystal, such as the solid state, the fibers and nanoparticles of this compound [4]. Tobermorite of 1.13 nm is most easily synthesized in mixtures with a molar ratio of CaO/SiO_2_ = 0.66–0.83 [5].

Xonotlite Ca_6_Si_6_O_17_(OH)_2_ is denoted by unique physical and chemical properties [6]. This mineral is formed in aqueous suspensions of CaO and SiO_2_ via hydrothermal synthesis at 200–350 °C [7]. Xonotlite fibers form needle-shaped crystals [8]. The main area of application is effective heat-resistant (up to 1000 °C) thermal insulating materials due to their high porosity, good fire resistance and strength [9]. Moreover, these xonotlite fibers are nanoscale in size, and, due to their developed specific surface, they can be used as catalysts in the production of hydrogen [10].

The formation of calcium silicate hydrates is inherently complex, and compounds are closely related to each other in their chemical composition and structure. The formation kinetics studies are further complicated by the co-formation of a semi-amorphous compound C–S–H(I) or metastable crystalline phases (Z-phase). For this reason, it is hard to obtain single-phase, pure and high-crystallinity degree materials [11].

The temperature of hydrothermal synthesis, its duration, the amount of solids in the suspension and its stirring speed determine the rate of 1.13 nm tobermorite formation, its degree of crystallinity and purity, the amount of intermediates and their stable existence. Temperature plays a crucial role. This compound takes hundreds of days to form at room temperature, yet in a high-temperature (175–200 °C) saturated water vapor environment, it forms in less than 24 h. Most of the work focuses on the 90–210 °C temperature range [12]. At about 200 °C (the exact value largely depends on the nature and amount of impurities), 1.13 nm tobermorite becomes metastable [13] and recrystallizes to xonotlite [14]. As a rule, the crystallinity of 1.13 nm tobermorite and the intensity of its diffraction peaks in X-ray curves increase with the increasing temperature. Moreover, the literature data indicates that the most favorable temperature for the formation of this compound is usually about 180 °C [14,15]. In it, sufficiently large crystallites of 1.13 nm tobermorite are formed at a competitive rate.

Lime, quartz sand and varieties of amorphous SiO_2_ are most widely used. The degree of 1.13 nm tobermorite crystallinity is also significantly influenced by the chemical composition, fineness and purity of the materials in use. This is especially important, however, when using natural raw materials or technogenic industrial products, as the impurities they contain can significantly adjust the synthesis process [16,17]. Moreover, 1.13 nm tobermorite is synthesized from many different CaO and SiO_2_-containing sources; lime, quartz sand and different varieties of amorphous SiO_2_ are the most widely used [18,19]. However, new sources of SiO_2_ are not limited to this; they tend to include silica fume *Elkem* (*Elkem Silicon Products*, Oslo, Norway), and *Grace Davison* (*Grace Davison Columbia*, Columbia, MD, USA), silica sand *Dorsilit* (*KRAFT Baustoffe GmbH*, Munich, Germany), and marble [20]. Research started looking for new raw materials which would contain a large amount of not only reactive SiO_2_ but also impurities that promote the formation of 1.13 nm tobermorite. Some non-traditional materials, such as kaolinite and metakoaline [21], K-feldspar [22], rocks trachyte [23], fuka [24], and opoka [25] are already being successfully used. The data provided in the literature shows that 1.13 nm tobermorite can be successfully synthesized from a variety of industrial technogenic products and even waste: coal and high alumina fly ash [12,26], biomass ash [27], blast furnace and steel slag [28,29], etc.

The formation of xonotlite largely depends on the composition of the initial mixture. Although some researchers [30] suggest that xonotlite can be synthesized from mixtures of very different basicity (molar ratio CaO/SiO_2_ = 0.41–1.66), most agree that hydrothermal reactions are most likely to occur when CaO/SiO_2_ = 1.0 [31]. The formation of xonotlite from lime–silica mixtures proceeds through the following intermediates: C–S–H(I), α-C_2_SH, and 1.13 nm tobermorite [32]. At the beginning of hydrothermal reactions, Ca-rich C–S–H gels were formed, while quartz did not completely react; then, 1.13 nm tobermorite began to form from these gels. The latter compound is not stable if there are free calcium ions in the reaction medium. The stoichiometric molar ratio is disturbed, and 1.13 nm tobermorite starts to recrystallize into xonotlite [33]. This conversion is topotactic, which means that the new compounds are formed without altering the morphology of the particles. As a result, xonotlite often slightly lacks Ca^2+^ ions, and its stability is achieved by Si–OH groups [34].

The main way to reduce the heat loss of buildings is through the use of suitable and high-quality insulating materials. By designing inorganic nanoporous products, their thermal resistance must be significantly increased [35]. One of the best materials of this type is silica aerogel [36], however, it is prohibitively expensive. Therefore, alternatives are needed, especially those that could use locally mined or otherwise sourced raw materials.

For this reason, it is worthwhile to further investigate insulating heat-resistant materials based on calcium silicate hydrates [37]. The properties of the products made from them are regulated by the *Standard EN 16977:2020* [38]. The classification of calcium silicate insulations is based on the compounds which form an insulation matrix. The crystalline phases can vary among 1.13 nm tobermorite, xonotlite and others, depending on the reaction conditions involved. The low resistance against temperature compared to other materials characterizes the heat-insulating products based on tobermorite. The most important performance aspects of xonotlite are as follows: low density and thermal conductivity; high mechanical strength; good heat resistance; low shrinkage up to 1050 °C; durability and resistance to chemical corrosion [9]. However, these properties can only be achieved by synthesizing compounds of high purity [39], i.e., with a minimum amount of semi-crystalline compounds that increase the shrinkage of the products.

The world produces a large number of good quality insulation materials, their range is wide, and all of them have their own areas of rational use and application. However, their nomenclature decreases when the operating temperature exceeds 500 °C, and silicate or ceramic materials are usually chosen here. In order to reduce the cost of products, it is best to use locally sourced raw materials. Unfortunately, they are often lacking and unsatisfactory in quality or price. For these reasons, it is often necessary to look for new, unconventional raw materials. However, for their successful use in the synthesis of calcium silicate hydrates, systematic studies of their properties and hydrothermal treatment parameters are required. Upon assessing this, the goal of this work is to synthesize 1.13 nm tobermorite and xonotlite from lime and opoka with a controlled structure and properties, and to evaluate the suitability of the obtained products for the production of heat-resistant thermal insulation materials.

## 2. Materials and Methods

### 2.1. Characterization of the Raw Materials

In this work, the following materials were used:

Lime (EN 459-1:2010) was sourced from *Lhoist Bukowa* sp., Poland. It was re-calcined for 1 h at 950 °C (the heating rate was equal to 8 °C/min; cooled naturally in an oven to 100 °C for 5 h, then transferred to a desiccator and cooled to room temperature) and ground in a ball mill for 30 min (the specific surface area by laser particle size analyser S_a_ = 650 m^2^·kg^−1^; CaO*_free_* = 99.12 mass%).

Opoka: (Stoniskiu–Zemaitkiemio quarry, Lithuania; SiO_2_ = 54.60 mass%; CaO = 22.10 mass%; Al_2_O_3_ = 2.53 mass%; Fe_2_O_3_ = 1.66 mass%; MgO = 0.55 mass%; K_2_O = 0.83 mass%). Opoka is mainly composed of opal silica and calcite, which makes it a highly suitable raw material and a source of both silica and CaO; more details can be found in Ref. [40]. Opoka was calcined at 775 °C for 1 h (the heating rate was equal to 8 °C/min; cooled in the same way as lime, only in 4 h) and milled in the ball mill for 30 min (S_a_ = 950 m^2^·kg^−1^).

### 2.2. Experimental Methods

#### 2.2.1. Preparation of Samples

The materials were mixed in a homogenizer *Turbula Type T2F* (Muttenz, Switzerland) for 1 h at 49 rpm.

Hydrothermal syntheses were carried out and stirred at 50 rpm (unless otherwise specified), 400 mL of suspension (W/S = 20.0) was placed in a *Parr Instruments*, model *4751*, (Moline, IL, USA), with a volume of 600 mL for autoclaving. The temperature of the saturated steam was 200 and 220 °C (when the heating rate was equal to 2 °C/min; cooling time to 100 °C for 2 h and 2 h 30 min, respectively), with the duration of isothermal curing of 4–72 h (Table 1). The synthesis products were filtered, rinsed with acetone to reduce carbonization, dried at a temperature of 100 ± 1 °C for 24 h, and passed through a sieve with an 80-μm mesh.

The samples (20 mm × 20 mm × 60 mm) were formed in a prefabricated stainless steel mold (60 mm × 70 mm × 60 mm) with two 20 mm high internal partitions (Figure 1). Many holes, 1 mm in diameter, were drilled at the bottom of the mold at a distance of 5 mm from each other. A water collection tank with a nozzle for connecting a vacuum pump was welded under it. The filter paper was placed on the bottom of the mold, the suspension was poured in, and vacuuming was initiated. As the mass began to dry, it was slowly pressed by the upper punch, also with holes 1 mm in diameter. It was pressed until the pressure reached 1.6 MPa and was maintained for 30 s without interrupting the vacuuming. The mold was dismantled, and the excess mass was cut off with a tightened wire at the top of the specimens. Three samples were formed simultaneously.

The hydrothermal curing of samples was carried out in a *Parr Instruments* pressure reactor, model *4555* (Moline, IL, USA), with a maximum pressure of 131 bar, the volume was 18.75 L, and the temperature range was from −10 to +350 °C. The temperature of the saturated steam was 200 and 220 °C (when the heating rate was equal to 1 °C/min; cooling time to 100 °C—5 h), and the duration of isothermal curing was 4–12 h.

#### 2.2.2. Instrumental Analysis

The specific surface area of the raw materials was determined by a laser particle size analyser *CILAS 1090 LD* (*Cilas*, Orléans, France) with a measurement range of 0.04–500 µm. The measurement principle is based on the diffraction of a light source by the sample. The results were processed by a PC running the Windows operating system. The dry method was used with an air supply of 6 bar.

The chemical composition analysis of the samples was performed by X-ray fluorescence spectroscopy (XRF) on a *Bruker X-ray S8 Tiger WD* (*Bruker AXS GmbH*, Karlsruhe, Germany) spectrometer (Rh tube; 60 keV). Powder samples compressed at 4 MPa were measured in helium atmosphere (software: *SPECTRA Plus QUANT EXPRESS,* Poulsbo, WA, USA).

The amount of free CaO was determined by following the Standard *ASTM C114–11b*.

The X-ray diffraction analysis (XRD) was performed on a D8 Advance diffractometer (*Bruker AXS GmbH*, Karlsruhe, Germany) operating at 40 kV; 40 mA; filtered with a Ni 0.02 mm filter to select the CuK*α* wavelength; scanned over the range 2*θ* = 3–70°; scanning speed 6°/min. The software *Diffrac.Eva v3.0* (*Bruker AXS GmbH*, Karlsruhe, Germany) was used for the identification of compounds and for the calculation of the size of crystallites. The size of crystallites of 1.13 nm tobermorite from the crystalline plane *h k l* (*d*–spacing 1.133 nm) and xonotlite (*h k l*; *d*–spacing 0.702 nm) was determined by following the Scherrer equation:(1)$$D_{hkl}=\frac{k_{hkl}\cdot \lambda }{\beta_{hkl}\cdot \cos\theta }$$
where *λ* is the wavelength of CuK*α* radiation, *θ* is the Bragg’s diffraction angle, *β_hkl_* is the full width at half maximum intensity, and *k* is a shape factor (the value used in this study was 0.94).

By employing the *Diffrac.Eva* software (Bruker AXS GmbH, Karlsruhe, Germany), we used the strategy of the entry points, the left angle and the right angle were used to measure the interval between two points and thus, obtain the intensity of the peaks. However, the highest value in the interval may be non-representative because of noise fluctuations. We determined the peak maximum by fitting a parabola through the points around the highest value and represented its value in scan units (by referring to it as plus *d* in Ä, because of applying the 2*θ* scan).

Simultaneous thermal analysis (STA; with a *Linseis PT1000* instrument; *Linseis Massgeraete GmbH*, Selb, Germany) was also employed for measuring the thermal stability and phase transformation: under N_2_ atmosphere at a heating rate of 10 °C·min^−1^; the temperature range was 30–1000 °C and crucibles of Pt were used.

Determination of the compressive strength was performed on a universal testing machine *FORM+TEST MEGA 10-400-50* (*FORM+TEST Seider&Co GmbH*, Riedlingen, Germany) at a loading rate of 0.5 kN/s.

Raw materials and samples were burned in a laboratory kiln *Nabertherm LV 15/11/P330* (*Nabertherm GmbH*, Bremen, Germany).

## 3. Results and Discussion

### 3.1. Hydrothermal Synthesis of 1.13 nm Tobermorite and Xonotlite

Our previous work [25] showed that products with a predominance of xonotlite in lime-calcined opoka mixtures cannot be obtained at a hydrothermal synthesis temperature of 180 °C. It is well known, however, that by increasing the temperature of the reacting medium and stirring it, the formation of crystalline calcium silicate hydrates is greatly accelerated [41]. For this reason, the temperature of the isothermal curing was 200 °C, and the suspension was stirred at 50 rpm. Therefore, it was determined that in the mixture with a molar ratio of CaO/SiO_2_ = 0.83, 1.13 nm tobermorite (PDF No. 04-011-0271; *d* = 1.133; 0.548; 0.308; 0.298; 0.282 nm) was dominating in the synthesis product as early as after 4 h of hydrothermal synthesis (Figure 2a, pattern 1). However, together with this compound, we identified the peaks of quartz (PDF No. 00-046-1045, *d* = 0.3343 nm) and calcite (PDF No. 04-023-8700, *d* = 0.3037 nm) as well. All the other minerals which are present in calcined opoka, such as cristobalite, tridymite and muscovite, react under these conditions.

The XRD results show that when prolonging the duration of hydrothermal synthesis, the intensity of the main 1.13 nm tobermorite peak (*d*-spacing—1.13 nm) increases slightly (Figure 2b). However, the peaks of quartz continued to decrease (Figure 2a), and, after 72 h of isothermal curing, only traces of the compound were left.

The results of simultaneous thermal analysis (STA) of the product obtained after 4 and 72 h of hydrothermal synthesis are presented in Figure 3. The DSC data within the 100–240 °C temperature range showed a very broad and vaguely expressed dehydration effect, which is related to the water removal from 1.13 nm tobermorite and semi-amorphous calcium silicate hydrate of the C–S–H(I) type, without a clearly expressed crystal lattice (Figure 3, curves 3 and 4). As mentioned in the introduction, this compound remains stable when the synthesis duration is insufficient, or when the molar ratio of the mixture is disturbed. Thermogravimetric analysis (TG) showed a slight difference in weight loss in the range of 100–240 °C—for the product synthesized for 4 h, it was equal to 6.32 mass%, and for the 72 h synthesized product, 6.23 mass%, and similarly, with a total weight loss of 12.1 mass% and 11.3 mass%, respectively (Figure 3, curves 1 and 2). In addition, the heat flow values of the exothermic effect at a temperature of ~840 °C (C–S–H(I) recrystallization into wollastonite) are also close: 12.5 mW·mg^−1^ of the synthesized for 4 h product (Figure 3, curve 3); 10.4 mW·mg^−1^ of the synthesized for 72 h product (Figure 3, curve 4). When summarizing these data, it can be stated that sufficiently pure 1.13 tobermorite from the lime–calcined opoka mixture at a temperature of 200 °C is already available in 4 h. Increasing the duration of the isothermal curing does not significantly affect the mineral composition of the product (either qualitatively or quantitatively). The small amount of quartz remaining in the product is likely to react during the treatment of the thermal insulation samples.

A negligible endothermal effect at 375 °C shows that the hydrogarnets (3CaO·Al_2_O_3_·(3−x)SiO_2_·2H_2_O, where x may vary from 0 to 3; PDF No. 04014-9841, *d* = 0.2722; 0.1975; 0.1688 nm) form at the beginning of the synthesis (Figure 3). This phenomenon is logical because opoka contains 2.53 mass% Al_2_O_3_. As the synthesis duration is prolonged, hydrogarnets decompose, and the released Al^3+^ ions intervene in the crystal lattice structure of 1.13 nm tobermorite. This sequence of reactions has been known for a long time [14,21]. The thermal effect in the DSC curve at a temperature of 664 °C is related to the decomposition of calcite, and this endothermic effect is very weak—the data of TG analyses showed less than 0.5 mass% mass loss (Figure 3, curve 1).

Dilatometric analysis of the powder sample, synthesized at 200 °C for 4 h and dried to a constant weight at 100 ± 1 °C, showed that the linear shrinkage in the temperature range from 25 to 700 °C is negligible and does not exceed 1.5 mass% (Figure 4).

Unfortunately, from 720 °C upwards, the sample begins to shrink very intensively, and the linear shrinkage in the 715–815 °C range exceeds 11 mass%. We assume that this is due to aluminum-containing impurities. As early as at the beginning of the synthesis, some quantity of Al^3+^ ions intervene in the C–S–H(I) structure and stabilize it, i.e., complicated recrystallization to the 1.13 nm tobermorite is observed. This phenomenon has long been studied and proven for calcium silicate hydrates of the same chemical composition, only with a higher degree of crystallinity, i.e., 1.13 nm tobermorite → xonotlite [33].

Thus, from the technological and economic point of view, the optimal synthesis conditions for 1.13 nm tobermorite precursor from which heat-resistant (up to 700 °C) thermal insulation materials can be produced are the following: molar ratio of lime–calcined opoka mixture CaO/SiO_2_ = 0.83, water–solids ratio W/S = 20, synthesis temperature—200 °C, duration—4 h, stirring intensity—50 rpm.

In order to produce heat-resistant (up to 1000 °C) thermal insulation products from the local raw material, opoka, it is necessary to study the parameters of xonotlite synthesis in detail.

The stoichiometry of xonotlite Ca_6_Si_6_O_17_(OH)_2_ corresponds to the molar ratio CaO/SiO_2_ = 1.0; therefore, the mixtures of this composition were studied. A temperature of 200 °C was chosen since this compound at 200 °C, although with difficulty, was nevertheless formed in unstirred lime–calcined opoka suspensions [42].

It was found, however, that in the product of 4 h hydrothermal synthesis, only 1.13 nm tobermorite formed from crystalline calcium silicate hydrates, however, the intensity of its diffraction peaks (Figure 5a, pattern 1) was much lower than in the product with CaO/SiO_2_ = 0.83. A small amount of xonotlite, together with a high crystallinity degree of 1.13 nm tobermorite, was found after 8 h of isothermal curing (Figure 5a, pattern 2). The intensity of the xonotlite main peak (*d*-spacing—0.702 nm) increased from 86 to 174 cps when the synthesis duration extended to 12 h (Figure 5b). The intensity of the 1.13 nm tobermorite peak (*d* = 1.13 nm) reached its maximum after 12 h of synthesis (1843 cps) and then began to decrease fairly rapidly to 1535 cps (24 h) and even plummeted to 981 cps (72 h). Although the intensity of the xonotlite peak was consistently increasing, even after 72 h, we still had a mixture of it and 1.13 nm tobermorite in the synthesis product (Figure 5).

The STA data of the products obtained after 12 and 72 h of synthesis at 200 °C under stirring at 50 rpm is presented in Figure 6. The endothermic effect at 204 °C indicates that 1.13 nm tobermorite is formed. The main exothermic effects were detected at 840 °C and 844 °C (Figure 6, curves 3 and 4). Moreover, at this thermal effect (related to C–S–H(I) recrystallization into wollastonite), the heat flow from 20.63 mW/mg (Figure 6, curve 3) decreased to 14.62 mW/mg (Figure 6, curve 4). When summarizing these results, it can be stated that the amount of semi-crystalline C–S–H(I) was lower after a longer duration of synthesis. This result is in good agreement with the XRD data, where the intensive peaks of xonotlite after 72 h of synthesis were identified (Figure 5a, curve 5).

Unfortunately, it has to be acknowledged that a product in which the dominant compound would be xonotlite could not be synthesized in this case, either. It is always formed with 1.13 nm tobermorite and C–S–H (I), and these calcium silicate hydrates significantly lower the application temperature of heat-resistant products.

Some authors suggest that the formation of xonotlite and other calcium silicate hydrates is accelerated by intensively stirring the suspensions [43,44]. However, the literature data on the positive influence of the mixing intensity on xonotlite synthesis is difficult to compare among alternative sources because of the different raw materials, equipment, and hydrothermal curing parameters being used. For these reasons, it is worth investigating in detail the influence of the stirring intensity on the formation processes of the target compounds. Therefore, hydrothermal synthesis at 200 °C for 12 h by stirring at 120–300 rpm when CaO/SiO_2_ = 1.0 was examined. Furthermore, it was determined that under stirring at 120 rpm, in the synthesis products, 1.13 nm tobermorite of the highest crystallinity degree (according to Taylor), together with xonotlite, is formed (Figure 7, pattern 1) [45]. A further increase in the stirring intensity had no effect on the mineralogical composition of the product (Figure 7).

However, we noticed that the stirring intensity during hydrothermal synthesis has a significant impact on the consistency of the suspension. Calcium silicate hydrates physically bind a large amount of water and form something like gels that arrange in the layers (sediments-water) very slowly. It has been observed that the thicker the suspension obtained during hydrothermal synthesis, the lower the density of the samples formed during simultaneous vacuuming and pressing. This parameter was named the *relative volume* and was determined by sedimentation. The synthesis products were cooled to 22 ± 1 °C, poured into a measuring cylinder, and kept at rest for 24 h. The relative volume of the synthesis product is the ratio of the volume of the sediments in a cylinder and their mass. This relative volume increases steadily with the increasing stirring rate (Table 2). We have found that when the value of this parameter is less than 10, samples with an average density of less than 200 kg·m^−3^ cannot be obtained or their strength is unsatisfactory. For this reason, the precursor for the formation of thermal insulation specimens was synthesized in stirred suspensions at 300 rpm.

By summarizing this part of the work, it can be stated, however, that at 200 °C, in the lime–calcined opoka mixture, we could not obtain products in which the predominant compound would be xonotlite. It is known, however, that by increasing the temperature of hydrothermal synthesis, the rate of xonotlite formation increases significantly [33]. Due to this reason, we performed hydrothermal synthesis at a temperature of 220 °C under stirring at 300 rpm. It was found that xonotlite was detected in the product as early as after 4 h of isothermal curing (Figure 8a, pattern 1) and its peak *d* = 0.702 nm intensity is higher than at 200 °C (Figure 8b). However, even after 72 h of curing, this compound coexists with 1.13 nm tobermorite (Figure 8a, curve 3).

Many researchers suggest that in order to obtain xonotlite via hydrothermal synthesis, mixtures with a molar ratio of CaO/SiO_2_ = 1.0 have to be used [31]. However, in the lime–calcined opoka suspension, in all the cases, the target compound was formed together with other low-base calcium silicate hydrates. This leads to the assumption that the molar ratio is too low for pure xonotlite formation. One of the reasons may be that not all CaO was involved in the formation of calcium silicate hydrates: part of it reacted with the impurities in opoka, and 2.95 mass% of CaO was in the composition of CaCO_3_, which was left after calcination of opoka at 775 °C. Increasing the calcination temperature is definitely problematic, however, because during the decomposition of carbonates, the obtained CaO is highly reactive, and, together with silica, wollastonite is formed. Due to this reason, the molar ratio of CaO/SiO_2_ in the reactive media can be lower than 1.0. Moreover, as mentioned in the introduction, 1.13 nm tobermorite is closely related to xonotlite by its structure and synthesis mechanism. Therefore, the lower molar ratio is more favorable for 1.13 nm tobermorite formation than it is for xonotlite.

Due to this reason, it was decided to increase the molar ratio up to CaO/SiO_2_ = 1.2 and investigate the formation of xonotlite in this particular case. It was determined, however, that under the same synthesis conditions, the reaction sequence in the products with this molar ratio changes significantly.

In summary, in the mixtures with CaO/SiO_2_ = 1.2, the formation of xonotlite at 200 °C is slow enough because, after 4 h of synthesis, only traces were identified (Figure 9, pattern 1). However, an important difference was observed in that 1.13 nm tobermorite is no longer formed in this mixture (Figure 10, patterns 2 and 3). In order to improve the formation of xonotlite, it was decided to increase the synthesis temperature.

The results of the XRD analysis of hydrothermal synthesis products from the mixture with CaO/SiO_2_ = 1.2 at 220 °C under 300 rpm stirring are presented in Figure 10. It was determined that the formation of xonotlite is extremely rapid, and as early as after 4 h of hydrothermal curing, it was identified as the main compound (Figure 10a, pattern 1). It should be noted, however, that in the synthesis product, we could not identify even traces of quartz. Furthermore, and most importantly, no other crystalline calcium silicate hydrates were found. The intensity of the xonotlite peak *d* = 0.702 nm was high—we did not obtain such values in the earlier stages of the work (Figure 10b). When prolonging the synthesis duration, no phase transformations were found to occur, and xonotlite remained stable for up to 72 h (Figure 10a). The only ongoing process was a steady increase in the intensity of the diffractive peaks of this compound (Figure 10b).

The exothermic effect at 842 °C in the DSC curve suggests, however, that in the 4 h synthesis product, there is quite a large quantity of C–S–H(I) (Figure 11, curve 3). However, the area of this effect in the 12 h synthesis sample decreased significantly, and only a blurred signal was captured (Figure 11, curve 4). Comparing the values of the heat flow, we see that they differ almost threefold, i.e., a decrease from 3.0 mW (4 h) to 1.03 mW (12 h).

Such results indicate that very small amounts of semi-amorphous compounds are formed under these conditions. According to literature data, the recrystallization of xonotlite to wollastonite is not associated with heat release or consumption [46]. Hence, this exothermic effect occurs only when semi-amorphous C–S–H(I) recrystallizes. This is in good agreement with the XRD data, as only intensive peaks of xonotlite were identified (Figure 10). In the DSC curve, we identified an exothermic effect at 845 °C, which suggests that xonotlite already dominates in the products obtained after 12 h of hydrothermal synthesis. In addition, in the DSC curve, we identified an endothermic effect at 723 °C (after 4 h), which is related to the decomposition of calcite. Moreover, the same effect is identified after 12 h of hydrothermal synthesis.

We would also like to mention that when the material under study contains an amorphous part, the XRD pattern of this material shows a visible blunt similar to a sloping hill. We did not notice this in the synthesis product pattern, therefore, we can say that it is composed entirely of a compound with a high degree of crystallinity, xonotlite, and a few percent impurities of semi-crystalline C–S–H (I) and calcite.

To summarize, the following technical parameters should be selected for the production of heat-resistant thermal insulating products with predominant xonotlite: the molar ratio of lime–calcined opoka mixture CaO/SiO_2_ = 1.2; water/solid ratio W/S = 20.0; the duration of hydrothermal synthesis to be set at 220 °C from 4 h to 12 h while stirring the suspension at 300 rpm.

### 3.2. Evaluation of Suitability of Opoka for the Production of Heat-Resistant Thermal Insulating Materials from Calcium Silicate Hydrates

As it can be seen from the data presented in Section 3.1, 1.13 nm tobermorite can be synthesized in a lime–calcined opoka suspension at a lower temperature and in a shorter time than xonotlite. However, its samples begin to shrink intensively from 720 °C (Figure 4); therefore, thermal insulation materials made of 1.13 nm tobermorite can only serve up to the above-mentioned temperature. On the other hand, there is a lot of heating equipment and pipelines in the industry with lower operating temperatures. Tobermorite-based products may be well suited for their insulation.

Precursors of thermal insulating materials were prepared from the suspension (CaO/SiO_2_ = 0.83, W/S = 20) which was obtained via hydrothermal synthesis at 200 °C for 4 h. The samples (20 mm × 20 mm × 60 mm) were formed in a vacuum-press form by using simultaneous water suction and pressing at 1.6 MPa. Afterward, they were reprocessed in an autoclave (200 °C, 4–12 h) and dried at 100 ± 1 °C until constant weight. The obtained results are presented in Table 3. It shows the mean values of the three samples; the scatter of the data did not exceed 10 mass%.

Medium quality products are obtained (average density ≤ 200 kg/m^3^, compressive strength ~0.9 MPa) that have their own areas of rational use. No changes in the qualitative mineralogical composition were observed in the XRD curves. We would like to point out, however, that a minor but steady increase in the strength values correlates well with the increase in the size of 1.13 nm tobermorite crystallites as the hydrothermal curing duration is prolonged. However, xonotlite should be used as a precursor to obtaining higher working temperature materials.

One of the essential requirements for heat-resistant thermal insulating products is their minimal shrinkage at high temperatures. Since semi-amorphous C–S–H(I) type calcium silicate hydrate does not have a clear crystalline structure, its shrinkage during recrystallization into wollastonite is very high. Meanwhile, the structure of the crystal lattice of xonotlite and wollastonite is very similar; therefore, the shrinkage during this recrystallization process is negligible. It means that the mineralogical composition of the products determines their working temperature. The change in the linear dimensions of the material and its ability to withstand the effects of a high temperature can be pre-determined by dilatometry analysis. When using this method, three samples of synthesized powder were investigated in order to select the optimal conditions for the production of precursors for thermal insulating products.

The linear shrinkage of the sample, which was synthesized at 200 °C for 72 h, is too high at 2.48 mass% (Figure 12, curve 1). Although we can see in the XRD curve that a high degree of crystallinity xonotlite has formed (Figure 9, curve 3), due to some amount of C–S–H(I) remaining in the product, items made from it can be used only up to 750–800 °C. By increasing the synthesis temperature to 220 °C, the shrinkage of the samples in the range of 30–1000 °C decreases significantly: after 4 h of isothermal curing, it is 1.68 mass%, and after 12 h, it is only 0.86 mass% (Figure 12, curves 2 and 3). This is in good agreement with the DSC data, which shows that the latter product is virtually free of C–S–H(I) type calcium silicate hydrate (Figure 11, curve 4).

Precursors of thermal insulating materials were prepared from the suspensions which were obtained via hydrothermal synthesis at 220 °C for 4, 8, and 12 h. It should be noted that the formed xonotlite physically binds a large amount of water, which results in very thick suspensions with a relative volume of about 20. The formed samples were autoclaved at 220 °C for 12 h and dried. As with the samples based on 1.13 nm tobermorite, no changes in the qualitative mineral composition were observed in the XRD curves. However, the size of the crystallites increases gradually, and, at the same time, the strength values of the samples increase (Table 4, columns 1–3). It is assumed that the xonotlite suspension should be synthesized at 200 °C for 8 h, as the average density of the obtained samples is ~180 kg/m^−1^, and the compressive and bending strengths exceed 1.5 and 0.8 MPa, respectively. By extending the synthesis duration to 12 h, the properties of the samples improve only slightly (Table 4, column 3).

By varying the duration of hydrothermal treatment at 220 °C from 4 to 8 h for the samples formed from the 8 h synthesized suspension, both their physical-mechanical properties and the size of the crystallites change only slightly (Table 4, columns 4–6). Thus, in the production of heat-resistant thermal insulation products based on xonotlite from the lime–calcined opoka mixture, the optimal technological parameters are as follows: molar ratio of the initial mixture CaO/SiO_2_ = 1.2; the water/solid ratio of the suspension W/S = 20.0; duration of hydrothermal synthesis at 220 °C—8 h, duration of autoclaving at 220 °C—4 h.

Linear shrinkage is only one of the indicators limiting the temperature of the use of products. Even more so at up to 800 °C, all calcium silicate hydrates are finally dehydrated and transformed to anhydrous forms, and in this case, mainly to wollastonite CaO·SiO_2_. During recrystallization, internal stresses may occur in the product and its strength may decrease. No less important is the decrease in strength during burning. To evaluate this property, the samples prepared under the above-outlined conditions were fired at 650, 800, and 1000 °C for 2 h. They did not show any cracks or other damage. After cooling to room temperature, the compressive strength of the samples was determined. The results are shown in Table 5. It can be stated, however, that even at a temperature of 1000 °C, despite losing crystalline water, the samples retain high strength for a sufficiently long time.

The thermal conductivity of the samples calculated according to the Japanese Standard *JISA1413* was relatively low at around 0.040 W/m·K. This is due to the fact that the xonotlite density is 2700 kg/m^3^, while the average density of the samples is ~180 kg/m^3^. This means that the solid material occupies only 4.8 mass% of the sample volume, while the remaining 95.2 mass% is air entrapped in very small pores.

In addition, the obtained compressive strength for xonotlite-based thermal insulation materials is sufficient for product transportation and installation on site.

## 4. Conclusions

The 1.13 nm tobermorite, with a high degree of crystallinity in the lime–calcined opoka mixture with a molar ratio of CaO/SiO_2_ = 0.83 at 200 °C, is formed rapidly at 4 h. However, even if the duration of the hydrothermal curing is extended to 72 h, a relatively high amount of semi-amorphous C–S–H (I) remains in the product. For this reason, tobermorite-based insulation products with good physical-mechanical properties (average density < 200 kg/m^3^, compressive strength ~0.9 MPa) but limited operating temperature (up to 700 °C) can be produced.The synthesis of xonotlite from industrial raw materials requires a very careful assessment of the amounts of impurities and their influence on the course of hydrothermal reactions. Impurities can react with CaO and alter the basicity of the reacting medium. In this case, the composition of the initial mixture needs to be adjusted.Sufficiently pure xonotlite must be synthesized from calcined opoka to produce heat-resistant materials. Even a small amount of C–S–H(I) remaining in the product increases the linear shrinkage of the samples when burned to 1000 °C from 0.86 to 2% or more. Xonotlite precursors physically bind a large amount of water, resulting in very thick suspensions with a relative volume of about 20. It has been observed, however, that the thicker the suspension obtained during hydrothermal synthesis, the lower the density of the formed samples.High-quality heat-resistant (up to at least 1000 °C) xonotlite-based products can be made from lime–calcined opoka mixtures. Our recommended parameters are as follows: a molar ratio of the initial mixture of CaO/SiO_2_ = 1.2; a water/solid ratio of suspension W/S = 20.0; the duration of hydrothermal synthesis to be set at 220 °C for 8 h and the duration of autoclaving to be set at 220 °C for 4 h. The relative volume of the suspension is ~20, the average density of the obtained samples is ~180 kg/m^3^, and the compressive strengths exceed 1.5 MPa.

## Figures and Tables

**Figure 1 materials-15-03474-f001:**
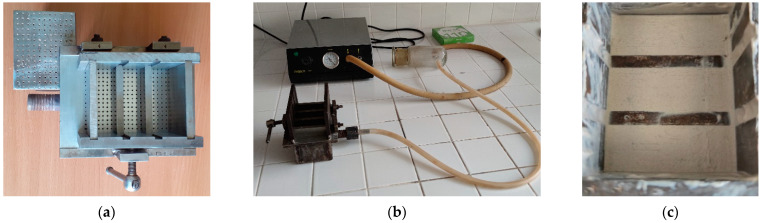
Mold of sample formation (**a**), formation process (**b**) and samples obtained (**c**).

**Figure 2 materials-15-03474-f002:**
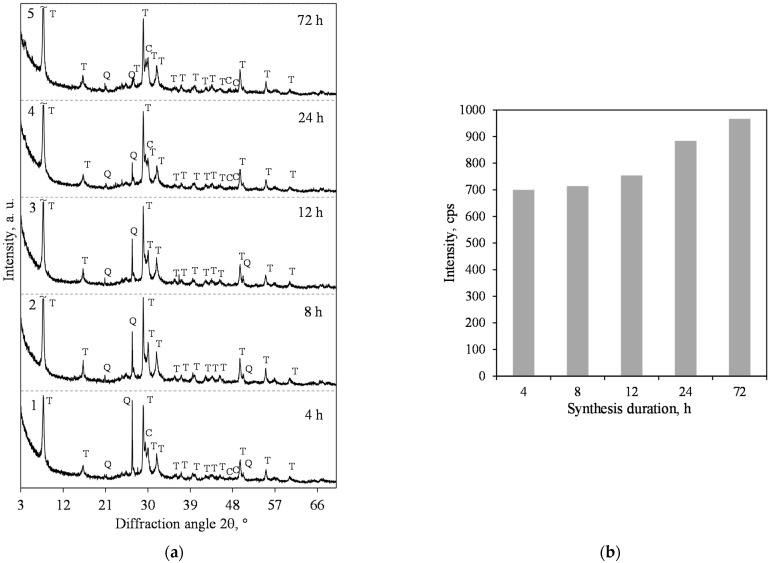
XRD patterns (**a**) and intensity of the 1.13 nm tobermorite (*d* = 1.13 nm) peak (**b**) of the hydrothermal synthesis products from lime–opoka mixture with CaO/SiO_2_ = 0.83 at 200 °C under 50 rpm stirring. Index: T—1.13 nm tobermorite, Q—quartz, C—calcite.

**Figure 3 materials-15-03474-f003:**
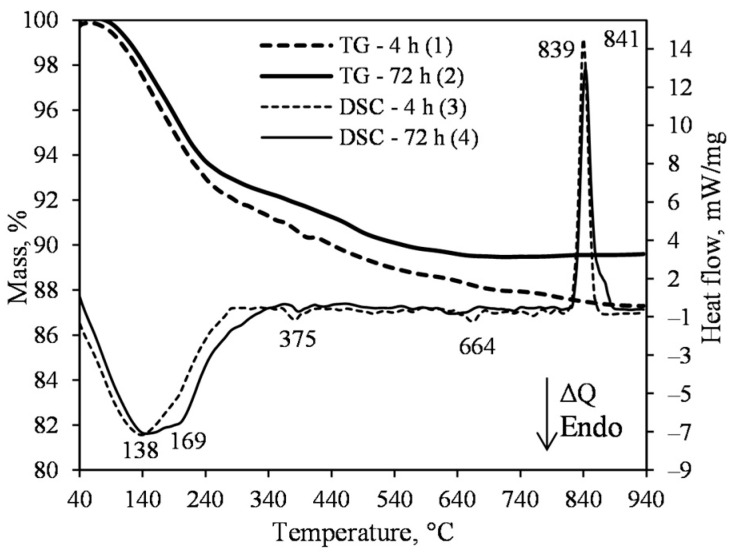
TG (1, 2) and DSC (3, 4) curves of the synthesis products from lime–opoka mixture with CaO/SiO_2_ = 0.83 at 200 °C under 50 rpm stirring after 4 h (1, 3) and 72 h (2, 4).

**Figure 4 materials-15-03474-f004:**
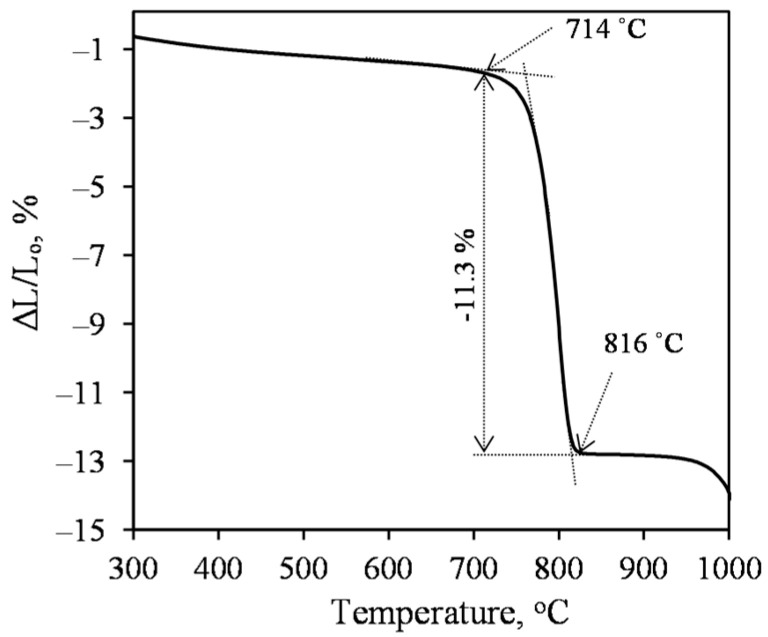
Dilatometric curve of the product from lime–opoka mixture with CaO/SiO_2_ = 0.83 at 200 °C under 50 rpm stirring after 4 h.

**Figure 5 materials-15-03474-f005:**
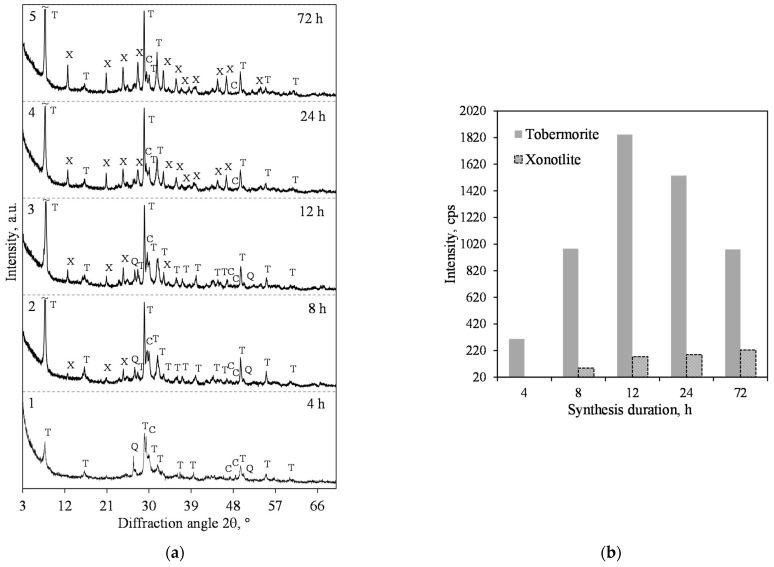
XRD patterns (**a**) and intensity of the 1.13 nm tobermorite (*d* = 1.13 nm) and xonotlite (*d* = 0.702 nm) peaks (**b**) of the hydrothermal synthesis products from lime–opoka mixture with CaO/SiO_2_ = 1.0 at 200 °C under 50 rpm stirring. Index: T—1.13 nm tobermorite, X—xonotlite, Q—quartz, C—calcite.

**Figure 6 materials-15-03474-f006:**
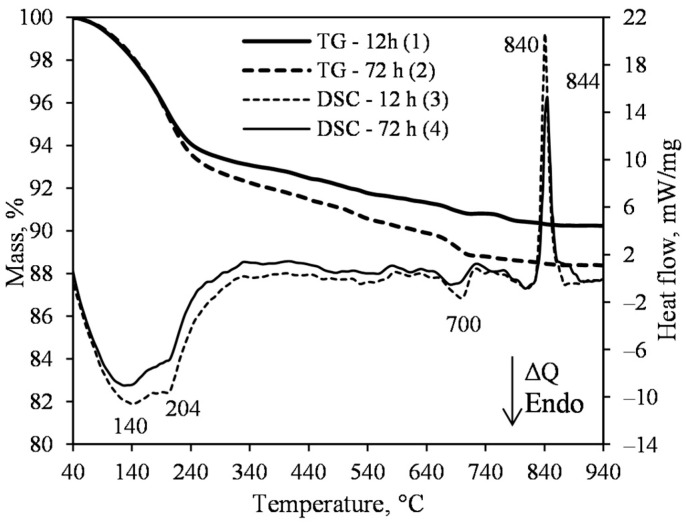
TG (1, 2) and DSC (3, 4) curves of synthesis products from lime–opoka mixture with CaO/SiO_2_ = 1.0 at 200 °C under stirring at 50 rpm after 12 h (1, 3) and 72 h (2, 4).

**Figure 7 materials-15-03474-f007:**
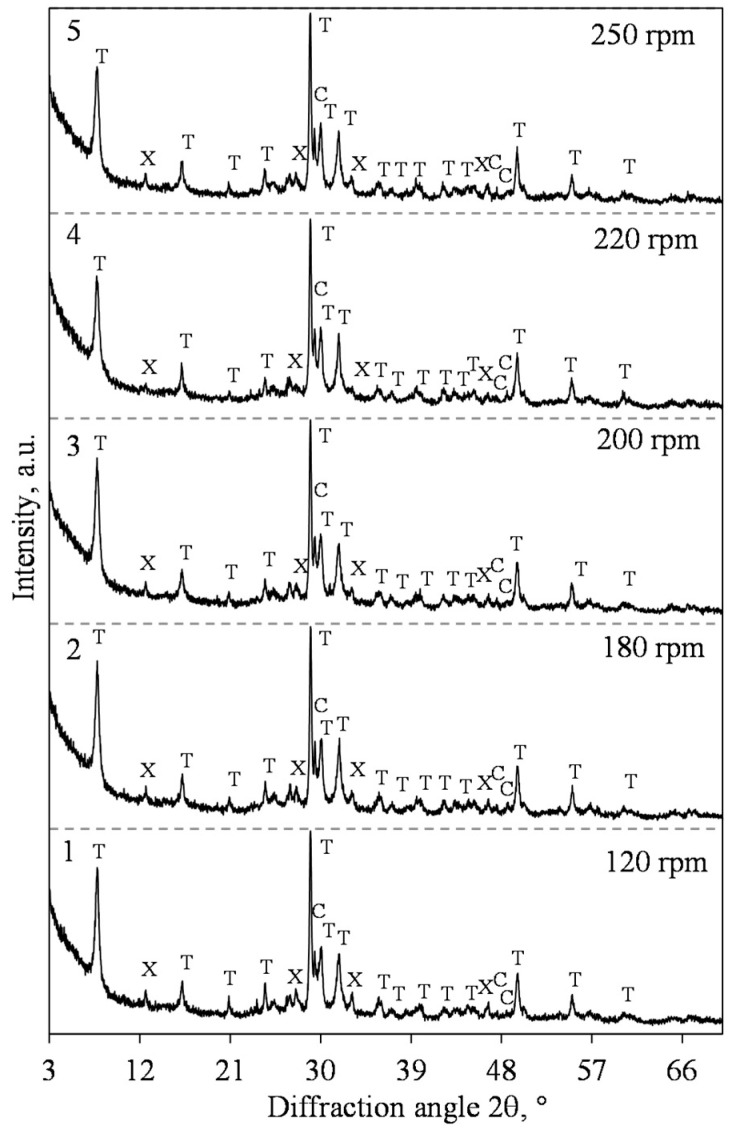
XRD patterns of hydrothermal synthesis products after 12 h under different stirring rates from lime–opoka mixture with CaO/SiO_2_ = 1.0 at 200 °C. Index: T—1.13 nm tobermorite, X—xonotlite, C—calcite.

**Figure 8 materials-15-03474-f008:**
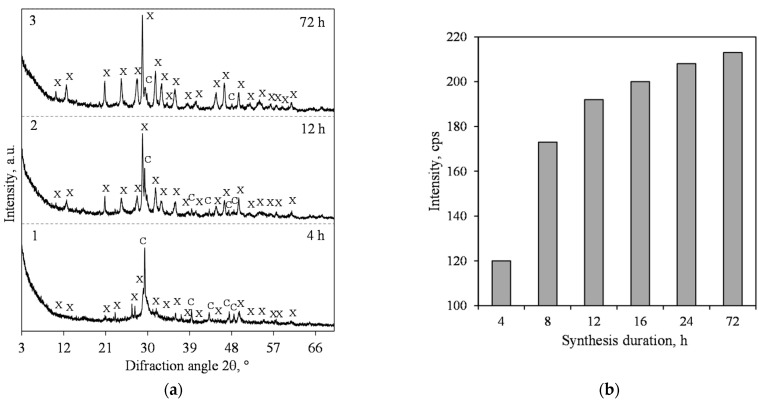
XRD patterns (**a**) and intensity of the xonotlite (*d* = 0.702 nm) peak (**b**) of hydrothermal synthesis products from lime–opoka mixture with CaO/SiO_2_ = 1.0 at 220 °C under 300 rpm stirring. Index: T—1.13 nm tobermorite, X—xonotlite, Q—quartz, C—calcite.

**Figure 9 materials-15-03474-f009:**
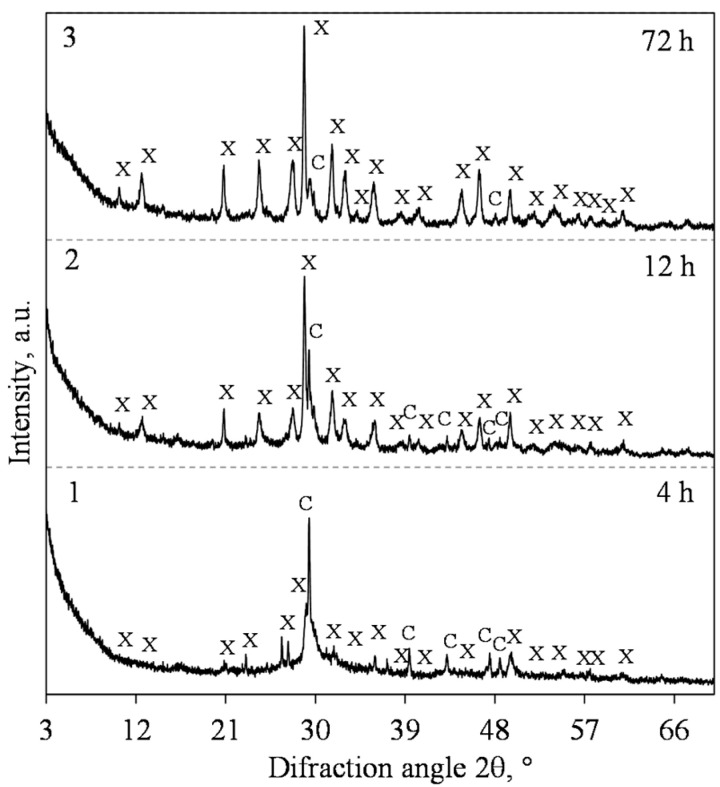
XRD patterns of hydrothermal synthesis products from lime–opoka mixture with CaO/SiO_2_ = 1.2 at 200 °C under 300 rpm stirring. Index: X—xonotlite, C—calcite.

**Figure 10 materials-15-03474-f010:**
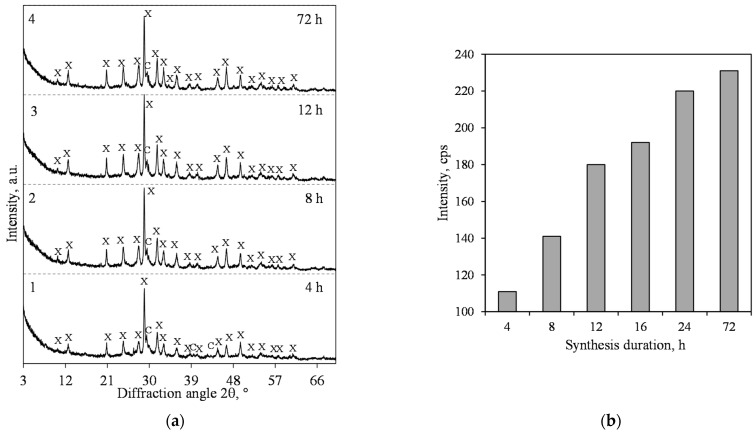
XRD patterns (**a**) and intensity of the xonotlite (*d* = 0.702 nm) peak (**b**) of hydrothermal synthesis products from lime–opoka mixture with CaO/SiO_2_ = 1.2 at 220 °C under 300 rpm stirring. Index: X—xonotlite, C—calcite.

**Figure 11 materials-15-03474-f011:**
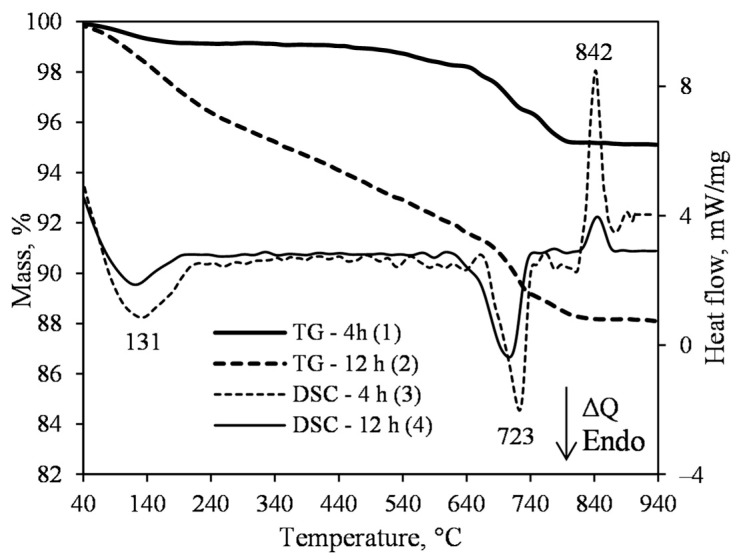
TG (1, 2) and DSC (3, 4) curves of the synthesis products from lime–opoka mixture with CaO/SiO_2_ = 1.2 at 220 °C under stirring at 300 rpm after 4 h (1, 3) and 12 h (2, 4).

**Figure 12 materials-15-03474-f012:**
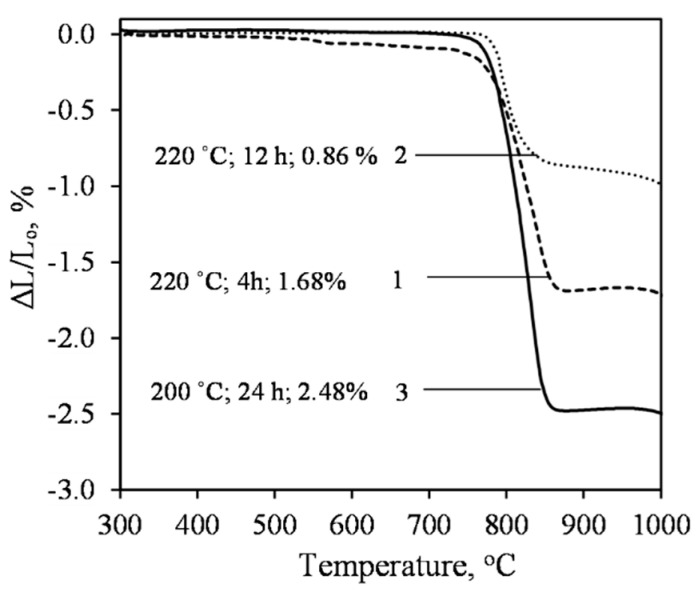
Dilatometry analysis curves of the synthesis products at 200 °C for 72 h (3) and at 220 °C for 4 h (1) and 12 h (2), when CaO/SiO_2_ = 1.2.

**Table 1 materials-15-03474-t001:** Main properties of hydrothermal synthesis.

System No.	Mixture with Molar Ratio CaO/SiO_2_	The Temperature of the Isothermal Curing, °C	Duration of Isothermal Curing, h	The Suspension Stirred Rate, rpm
1	0.83	200	4; 8; 12; 24; 72	50
2	1	200	4; 8; 12; 24; 72	50
3	1	200	12	120; 180; 200; 220; 250;300
4	1.0	220	4; 8; 12; 16; 24; 72	300
5	1.2	200	4; 12; 72	300
6	1.2	220	4; 8; 12; 16; 24; 72	300

**Table 2 materials-15-03474-t002:** Ratio of the volume of sediments and their mass after 24 h of sedimentation.

Stirring rate, rpm	50	120	180	200	220	250	300
Relative volume, cm^3^/g	6	7	7.5	10	11	13.5	15

**Table 3 materials-15-03474-t003:** The main properties from 1.13 nm tobermorite-based samples.

Properties	Duration of Hydrothermal Synthesis at 200 °C—4 h	Curing Duration at 200 °C, h			
		4	6	8	12
Size of crystallites, nm	35.0	38.1	39.9	43.1	50.3
Average density, kg/m^3^	–	193.3	188.5	191.8	190.1
Compressive strength, MPa	–	0.77	0.87	0.92	0.93
Bending strength, MPa	–	0.46	0.56	0.53	0.59

**Table 4 materials-15-03474-t004:** Main properties from xonotlite-based samples.

Properties	Duration of Synthesis at 220 °C, When Curing Duration at 220 °C—12 h			Curing Duration at 220 °C, When Duration of Synthesis at 220 °C—8 h		
Columns number	1	2	3	4	5	6
Hydrothermal treatment, h	4	8	12	4	6	8
Size of crystallites, nm	30.4	31.7	34.0	27.3	27.9	30.7
Average density, kg/m^3^	214.2	184.0	178,1	178.6	181.3	183.1
Compressive strength, MPa	1.40	1.54	1.84	1.40	1.34	1.44
Bending strength, MPa	0.78	0.83	0.86	0.61	0.68	0.85

**Table 5 materials-15-03474-t005:** Compressive strength of burned xonotlite-based samples.

Properties	Burning Temperature, °C, With Duration of 2 h		
	650	800	1000
Compressive strength, MPa	1.39	1.22	0.97
Variation, % of initial value	96.5	84.7	67.3

## Data Availability

The data supporting the findings of this study are available from the corresponding author, R.S., upon reasonable request.

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
