# Peer review of "Synthesis of High Crystallinity 1.13 nm Tobermorite and Xonotlite from Natural Rocks, Their Properties and Application for Heat-Resistant Products"

_materials, 2022, doi:10.3390/ma15103474_

Round 1

Reviewer 1 Report

Recommendation: not recommended

Comments:

The topic, i.e. mineral binder chemistry and processes, is extremely timely and important. Deep lacks exist in the methods and the evaluation of experimental data. The writing and the way the data are presented and discussed require improvement. For the moment, this manuscript cannot be recommended for publication.

Synthesis is always interesting for publication. However, it is only of value if the description given in ‘Methods’ allows the reader to reproduce the work. This point is not fulfilled in the submitted version. The problems start in the first sentence, where a temperature is given, but neither a heating nor a cooling rate. Furthermore, to the best of my knowledge, a ball mill does not measure the surface area. So how should the reader know from that sentence - “ground in a ball mill until reaching … surface area X” - how long to mill the material? These deep-going problems make it, for the moment, impossible on the one to reproduce this synthesis and on the other to seriously judge the products.

However, the authors report about a certain factor, like the relative volume, influencing the described synthesis. The relative volume seems to be key to the success of the xonotlite synthesis but is not named in the recommendation of parameters in conclusion. Furthermore, the parameters and their influence on the synthesis is not explained in the paper.

The XRD only represents the crystalline part of a material. How much material is used in one charge, and what are the yields of the synthesis?

The silicates produced here belong to the family of CSH phases, and all of them have a temperature limit above which they which lose their water and turn into CS phases. But this step happens only one time, so for the industry it could be a final step in the synthesis of the heat resistant material.

Author Response

Please note that the numbering of figures and tables has been changed and is indicated in our replies. Please see the attachment.

Reviewer 2 Report

The manuscript entitled “Synthesis of High Crystallinity 1.13 nm Tobermorite and Xonotlite from Natural Rocks, Their Properties and Application for Heat-Resistant Products” requires minor revision before publication. The author mainly described the crystal formation of 1.13 nm tobermorites and xonotlites based on their previously works on references 25, 41, and 42. The works in the thermal analysis and XRD are sound to clarify T, Q, X, C phases depending on the CaO/SiO2 ratios. Although the manuscript is ready to be accepted, several suggestions were made to authors for revision. Please find the comments below.

(1) The English usage in the manuscript requires improvements or suggestions from native speakers. Many sentences are confused, for example, at the line 62-63.

The chemical composition, purity and dispersion of the raw materials have no less 62 impact on the rate of formation, the mineral composition and the crystallinity degree of 63 the resulting products.

(2) Some paragraph goes to nowhere in the introduction. It does not relate to the main idea of the article. Please consider the paragraph between line 91 and line 95.

(3) At line 226 and 227, the unit isn’t unified to present the concept of mass loss. 6.32% ---> -6.23; -12.1% ---> 11.3%

(4) There are many diffraction peaks in the Figure 1a, 4a, 6, 7a, 8, and 9a for T, Q, X, C phases after hydrothermal synthesis. Please identify which peak was used to present the corresponding intensity of crystals 1.13 nm tobermorite or xonotlite. How do authors plot Figure 1b, 4b, 7b, and 9b?

(5) Please separate Figure 2 and Figure 3 independently.

(6) The format of Table 4 is different to others. It should be unified.

(7) References are out of date. The latest one is ref 38 Standard EN 16977: 2020.

Author Response

(The authors gave the same response as above.)

Reviewer 3 Report

This research follows on from a previous study on the synthesis of tobermorite at 180 °C from opoku (DOI: 10.1007/s10973-018-7418-1) by the same group. Novel data are presented and this paper will appeal to chemists, materials scientists and engineers with an interest in the hydrothermal synthesis and characterisation of calcium silicate hydrates.

The English language and grammar require modest revisions at the copy editing stage.

The abstract is comprehensive, concise and informative.

In general, the introduction provides a thorough and relevant overview of the hydrothermal chemistry of tobermorite and xonotlite.

Line 51: The authors state, ‘The temperature increase had a significant impact on the reaction kinetics; the synthesis duration decreased from a hundred days at room temperature to mere hours.’. The authors must inform the reader of this elevated temperature.

Line 70: The full commercial names, the cities and countries must be provided for the chemical companies listed in Line 70.

Experienced researchers would welcome a more explicit account of the objectives and experimental work at the end of the Introduction to enable them to progress directly to the Results and Discussion without having to formally consult the Materials and Methods section. Although, I don’t insist on this.

In section 2.2.1, the authors should provide photographic images of the steel mold to accompany the description (since this is not standard equipment).

In section 2.2.1, the authors should provide a table of the mix components and reaction parameters for each synthesis.

Can the authors please provide quantitative Rietveld analysis to determine the actual quantities of phases present and the total crystallinity of their hydrothermal products? This information would add value to the manuscript.

I have no confidential comments for the editors.

Author Response

(The authors gave the same response as above.)

Round 2

Reviewer 1 Report

The authors have answered most of my questions and revised the manuscript consequently. One important question has not been answered: As the XRD only represents the crystalline part of a material, what are the yields of these synthetic routines? Additionaly, the methodic part is still not finished. For example: % is not clear was is meant - atm%, V%, mass%?

If they finally polish these things, the manuscript can be published.
